# *BRCA*-Mutated Pancreatic Cancer: From Discovery to Novel Treatment Paradigms

**DOI:** 10.3390/cancers14102453

**Published:** 2022-05-16

**Authors:** Naomie Devico Marciano, Gianna Kroening, Farshid Dayyani, Jason A. Zell, Fa-Chyi Lee, May Cho, Jennifer Goldstein Valerin

**Affiliations:** Department of Medicine, University of California, Irvine, CA 92868, USA; ndevicom@hs.uci.edu (N.D.M.); gkroenin@hs.uci.edu (G.K.); fdayyani@hs.uci.edu (F.D.); jzell@hs.uci.edu (J.A.Z.); fachyil@hs.uci.edu (F.-C.L.); mayc5@hs.uci.edu (M.C.)

**Keywords:** pancreatic cancer, *BRCA*, chemotherapy, DNA repair, PARP

## Abstract

**Simple Summary:**

Approximately 10–20% of pancreatic cancer patients will have a mutation in their DNA, passed on in families, that contributes to the development of their pancreatic cancer. These mutations are important in that they effect the biology of the disease as well as contribute to sensitivity to specific treatments. We describe the critical role that these genes play in various cellular processes in the body that contribute to their role in cancer development and normal cellular function. In this review, we aim to describe the role of certain genes (BRCA1 and BRCA2) in the development of pancreatic cancer and the current and future research efforts underway to treat this subtype of disease.

**Abstract:**

The discovery of *BRCA1* and *BRCA2* in the 1990s revolutionized the way we research and treat breast, ovarian, and pancreatic cancers. In the case of pancreatic cancers, germline mutations occur in about 10–20% of patients, with mutations in *BRCA1* and *BRCA2* being the most common. *BRCA* genes are critical in DNA repair pathways, particularly in homologous recombination, which has a serious impact on genomic stability and can contribute to cancerous cell proliferation. However, BRCA1 also plays a fundamental role in cell cycle checkpoint control, ubiquitination, control of gene expression, and chromatin remodeling, while BRCA2 also plays a role in transcription and immune system response. Therefore, mutations in these genes lead to multiple defects in cells that may be utilized when treating cancer. *BRCA* mutations seem to confer a prognostic benefit with an improved overall survival due to differing underlying biology. These mutations also appear to be a predictive marker, with patients showing increased sensitivity to certain treatments, such as platinum chemotherapy and PARP inhibitors. Olaparib is currently indicated for maintenance therapy in metastatic PDAC after induction with platinum-based chemotherapy. Resistance has been found to these therapies, and with a 10.8% five-year OS, novel therapies are desperately needed.

## 1. Introduction

Genetic mutations have many causes, including DNA replication errors, exogenous and/or endogenous mutagen exposure, enzymatic modifications of DNA, or defects in the DNA repair machinery [1]. When these mutations occur in essential genes, they may contribute to the development of cancer. Cancer-causing genetic mutations can either be inherited or acquired over the lifetime of an individual.

The discovery of BReast CAncer gene 1 (*BRCA1*) and *BRCA2* in the 1990s revolutionized the way we research and treat breast, ovarian, and pancreatic cancers [2,3]. Since their discovery, their mechanisms of action have been extensively studied in the hopes of elucidating how these proteins function as tumor suppressors and their role in DNA repair. *BRCA* genes are most thought of as tumor-suppressor genes that are inherited in an autosomal-dominant fashion with incomplete penetrance [4]. Losing the function of tumor-suppressor genes is catalytic in the chain of events that drives tumorigenesis. Both *BRCA* genes play important roles in the transcriptional regulation of gene expression as well as in the recognition and repair of DNA damage, notably double-stranded breaks (DSB) as a part of homologous recombination [5]. Mutations in *BRCA1* and *BRCA2*, which impair their normal function, have been associated with an elevated risk of breast and ovarian cancer, as well as an increased risk of pancreatic, prostate, stomach, and many other cancers. *BRCA2* mutations are found in up to 5.7% of pancreatic adenocarcinomas and mutations in *BRCA1* are found in 2.4% [6]. Among pancreatic cancer patients of Ashkenazi Jewish descent, *BRCA* mutations are found in up to 18% of individuals [6]. Though *BRCA1* and *BRCA2* are associated with cancer risk, tumors that are identified as having “BRCAness” appear to be susceptible to specific therapeutic approaches [7]. “BRCAness” is used to describe cases in which homologous recombination repair (HRR) defects exist in a tumor in the absence of a germline *BRCA1* or *BRCA2* mutation [8]. Targeting DNA damage repair (DDR) pathways provides an opportunity to induce synthetic lethality in *BRCA-mutated* cancer cells [9]. The use of DNA-damaging chemotherapies such as platinum agents [10], as well as treatment of *BRCA-*deficient tumors with Poly adenosine diphosphate (ADP)-ribose polymerase (PARP) inhibitors, have proven to be highly tumor-specific treatments, indicating a predictive role for *BRCA* mutations in the treatment of PDAC [9]. In addition, *BRCA* mutations have been correlated with overall better survival in PDAC, indicating prognostic utility [11].

The potential treatment implication for *BRCA1* and *BRCA2* mutation-positive tumors has generated a multitude of studies in the different pathways modulated by these genes. *BRCA1* is instrumental in the DDR of DSB at multiple levels, it is involved in sensing the broken ends, it acts as a mediator for the formation of the DNA repair complex, and it also signals the activation of cell cycle checkpoints [6]. BRCA2 differs from BRCA1 in that it does not display such a large range of functions. BRCA2 has been primarily associated with HRD and with the tumor suppressive function of this repair process [6,12,13]. The critical functions of BRCA1 and BRCA2 in the repair mechanisms and genome stability have made these genes the target of decades of investigations in the hopes that elucidating their mechanisms can provide new avenues for cancer treatments.

## 2. Discovery of *BRCA1* and *BRCA2*

In the early 1980s, 1579 breast cancer patients were interviewed by the NCI’s Surveillance, Epidemiology, and End Result Program to determine if other members of their family had breast or ovarian cancer. In a study published in May 1988, Dr. Mary-Claire King found that familial clustering of breast cancers could be explained by inheriting an autosomal-dominant, susceptibility allele, which she estimated affected 4% of families in the study [14]. Efforts were then geared towards locating this gene.

In 1990, as the human genome project was taking off, the existence of *BRCA1* was proven by mapping predisposition to young-onset breast cancer families. In 1994, a team led by Mark Skolnick and Myriad Genetics sequenced *BRCA1* [15]. In a follow-up study that year, it was revealed that there existed multiple mutations that predispose carriers to cancers, including: a 1 base pair insertion, a premature stop codon, an 11 bp deletion, a missense mutation, and a putative regulatory mutation [1]. That same year, using a similar technique of linkage analysis, *BRCA2* was mapped to chromosome 13, launching a second race to clone this gene. In 1998, Myriad Genetics was awarded a patent for both *BRCA1* and *BRCA2* genes and began offering commercial testing for these hereditary forms of cancer.

In 2013, the Supreme Court of the United States declared Myriad’s patents on isolated genes which they had obtained in 1998 to be invalid. This was an important ruling as it allowed other genetic testing companies to begin offering genetic testing at lower rates, and it also opened the way for new research into treatments [16].

While the field of breast and ovarian cancer was advancing through the discovery of *BRCA* genes, family history of pancreatic cancer was also recognized in the late 1960s [17]. Through multiple population-based case-control studies, it was shown that compared to population controls, individuals with a first-degree relative (FDR) with pancreatic cancer have a 3.2-fold increased risk of developing pancreatic cancer [18]. Particularly, germline *BRCA2* mutations have been associated with an increased risk of pancreatic cancer (RR = 3.51; 95% CI = 1.87–6.58) [19]. Studies have shown that in patients with pancreatic cancers with 2 or more FDR with pancreatic cancer, the prevalence of *BRCA2* mutations ranges from 17% to 19% [20,21]. Furthermore, Goggins et al. demonstrated that in patients without a family history of pancreatic cancer, 7.3% had germline *BRCA2* mutations [22]. There is large heterogeneity of *BRCA1* and *BRCA2* mutations as well as phenotypic heterogeneity between family lineages that share common mutations [5].

*BRCA1* is located on chromosome 17q21 and plays several roles in the maintenance of genetic stability in proliferating cells. The BRCA1 protein consists of 1863 amino acids [1,23]. The C-terminus contains what is known as the BRCT (BRCA1 C Terminus) domain, which is recognized by multiple DNA repair proteins. It is understood to be a protein–protein interaction site to build a large protein complex of up to 3 MDa with other DNA repair proteins (BRIP1, CHK1, TOPBP1, ATP) and take part in cell cycle checkpoint control [24,25,26]. BRCA1 also has a coiled-coil (CC) domain for PALB2 binding. In its N-terminus, there is a structurally conserved RING-finger domain involved in protein ubiquitination. This RING-finger domain is a zinc-binding motif which can bind BARD1 and BAP1 [27,28]. BRCA1 forms a heterodimer complex with BARD1, which results in E3 ubiquitin ligase activity [29]. *BRCA1* is also known to play a role in the response to DNA damage via its BRCT domain [30]. Lastly, BRCA1 also interacts with BRCA2 though the bridging protein PALB2 (partner and localizer of BRCA2), facilitating the RAD51 filament formation.

*BRCA2* is located on chromosome 13q12-13 and is 3418 amino acids long. *BRCA2* is characterized by a very large exon 11 containing eight BRC peptide motifs, which play a key role in BRCA2’s interaction with RAD51. Through its interaction with BRCA2, RAD51 can overcome the inhibitory effect of the high-affinity ssDNA binding protein RPA which coats ssDNA [30]. BRCA2 also preferentially binds ssDNA, and this promotes the assembly of RAD51 onto ssDNA over dsDNA, a step that is critical for the invasion of a homologous DNA [30]. In its C terminus, BRCA2 has a ssDNA binding domain. In its N-terminal, BRCA2 possesses a PALB2 interaction domain.

## 3. Predisposition to Pancreatic Cancer with *BRCA* and Associated Gene Mutations

*BRCA1* and *BRCA2* are known as tumor-suppressor genes that function to limit inappropriate cell growth and signal cell death when needed. Furthermore, *BRCA* genes are key players in DNA repair which prevent the accumulation of mutations in other cancer-related genes [31]. Therefore, a loss of function of tumor-suppressor genes can have a serious impact on genomic stability and can contribute to cancerous cell proliferation.

Cancers can arise from two distinct types of mutations, somatic and germline. Most cancers (75–80%) are sporadic, with somatic gene mutations or other genomic alterations caused by exposure to UV radiation, chemical exposure, or infectious agents, that occur over a person’s lifetime. These mutations have allowed for cancer acquisition or progression and are found only in tumor cells [32]. Conversely, germline mutations are inherited mutations that occur in germ cells and are therefore present in all cells of the body. It is estimated that 5–10% of all cancers are caused by inherited genetic mutations [33]. An important distinction is that hereditary cancers involve a genetic mutation that has been passed down from generation to generation, whereas familial cancers do not appear to be caused by genetic mutations in a single gene. Instead, it is believed that familial cancers result from multiple influences, such as a combination of multiple genes and factors such as diet, exercise, and shared environmental factors, among others [32]. Of note, individuals with familial cancers tend to develop these at later stages of life and may have multiple family members with multiple cancers [32]. On the other hand, some clues suggesting hereditary cancers include a young age of diagnosis (usually before age 50), multiple family members with the same or related type of cancer, the cancer tends to develop in multiple sites in the body, and rare cancers such as male breast cancer can occur [32].

In the case of pancreatic cancers, recent studies reported that germline mutations in genes such as *BRCA1, BRCA2, PALB2,* and *CDKN2A* occur in about 10–20% of patients without extra-pancreatic manifestations, and 5–8% of patients with pancreatic cancer without family history are in fact carriers of germline mutations [34,35]. Some studies have reported a near doubling of the risk of pancreatic cancer in *BRCA* female carriers [36]. The prevalence of *BRCA2* mutations in hereditary pancreatic ductal adenocarcinoma (PDAC) in non-Jewish, ethnically diverse populations has been reported to range from 6% to 17% [37]. In the Ashkenazi Jewish population, for example, there exist three predominant deletion mutations that have been detected in a majority of PDAC high-risk families in *BRCA1* (185delAG, 5382InsC) and *BRCA2* (6174delT). Other mutations that account for 10% of familial susceptibility to pancreatic cancer include mutations in *STK11, PALB2, CDKN2A, ATM, TP53, MLH1, MSH2, MSH6, PMS2,* and *EPCAM*. Mutations in the *STK11* gene (Peutz-Jeghers) represent a 130-times increased risk of pancreatic cancer relative to the general population [20,38]. *PALB2* gene mutation represents a 3.5–6.2 times increased risk [39]. Mutations in the *CDKN2A* gene, which are linked to the familial atypical multiple mole-melanoma (FAMMM) syndrome, are associated with a 13- to 22-fold increased risk of pancreatic cancer [40]. Mutations in the mismatch repair genes *MLH1, MSH2, MSH6,* or *PMS2,* which are associated with Lynch Syndrome, represent an 8.6-fold increased risk of pancreatic cancer [41]. A recent study found that the increased risk of developing pancreatic cancer with *ATM* mutations is age-dependent and ranges from 0.08% at age 30 to 9.53% by age 80 [42].

These observations have proven that it is critical that patients with pancreatic cancers are tested for germline mutations. Traditionally, germline testing in pancreatic cancer was performed only if the patient presented with criteria associated with one of the known cancer syndromes associated with increased pancreatic cancer risk. In 2018, the NCCN proposed that all patients with pancreatic cancer undergo germline genetic testing for the following predisposition genes: *BRCA1, BRCA2, PALB2, CDKN2A, ATM, TP53, MLH1, MSH2, MSH6, PMS2, STK11*, as well as *PRSS1/SPINK1* and *CFTR,* if the clinical history is suggestive of hereditary pancreatitis or cystic fibrosis, respectively [35]. The identification of such mutations would provide potential routes for the treatment of PDAC, as treatment choices are frequently stratified based on these mutations, as well as indicate the need for genetic counseling of family members. Furthermore, somatic testing might also be performed to identify how the tumor behaves and the risk of interaction with a germline mutation.

## 4. DNA Repair Mechanisms Compromised by *BRCA1* and *BRCA2* Mutations

According to the Knudson hypothesis, also known as the two-hit hypothesis, in the case of most tumor-suppressor genes (such as *BRCA1* and *BRCA2*), both alleles need to be inactivated to cause a phenotypic change [31]. In a heterozygous individual with a germline mutation, the inactivation of the second wild-type allele of a tumor-suppressor gene is termed loss of heterozygosity mutation. This loss of heterozygosity mutation can be caused by point mutations or small deletions, chromosomal deletions, or breaks [31]. DNA damage may lead to a high risk of tumorigenesis if it is not properly repaired.

DNA stability is largely compromised when DSBs occur. It is therefore essential that cells have multiple pathways available to repair DSBs. The dominant pathways include homology-directed repair (HDR), nonhomologous end joining (NHEJ), and microhomology-mediated end joining (MMEJ). Of these, HDR is more accurate as it uses the sister chromatid as a template to return the DNA sequence to its original form. Failure or defects in HDR are troublesome as they allow for the accumulation of mutations and ultimately, genomic instability. When HDR repair is impaired by *BRCA* mutations, for example, DNA repair is achieved by non-conservative forms, such as non-homologous end joining. Whereas HRD has high fidelity, in NHEJ, the ends of the break are not resected before being rejoined, which can induce deletion and insertion during this repair mechanism. These DNA modifications, particularly the DNA deletions, can occur in crucial cancer genes and aggravate the tumorigenesis effect of *BRCA* mutations.

Following a DSB, the HDR machinery is activated. The first crucial step in HDR is the generation of a 3′ single DNA strand by resection of the 5′end. End-resection involves MRN and CtIP proteins. The MRN complex has dual functions in sensing and signaling the DSB, and it is also required in all phases of cell cycle checkpoint signaling [43]. The BRCA1–CtIP complex promotes CtIP-mediated 5′-end resection of DSB [6]. It has been suggested that BRCA1 plays a role in the initial step of end resection by competing against 53BP1(an NHEJ factor) [30,44]. It is worth pointing out that HDR is promoted during the S/G2 phase of the cell cycle, and that it occurs in a cellular milieu that also supports NHEJ [30,44]. Therefore, end resection is a major determinant of whether NHEJ or HDR will be used to repair the DSB. End resection occurs in two steps: initially, a short (<100 bp) 3′ overhang is formed, then in a second phase, the 3′ end overhang is extended. RAD51 is recruited to bind to these overhangs. This is achieved through the interaction of BRCA1, PALB2, and BRCA2 (Figure 1).

This interaction occurs when BRCA1 binds to PALB2, which recruits BRCA2 to the complex (Figure 1). BRCA2 mobilizes the formation of RAD51 filaments on the 3′single strands, as well as a strand invasion on the sister chromatid which initiates the repair of the DNA break by DNA synthesis. BRCA2 binds RAD51 through the interactions of the BRC repeat regions encoded by exon 11. Of the eight BRC repeats, BRC3 and BRC4 have the strongest interaction with RAD51 [45]. BRCA2 aids in the formation of RAD51 filaments at two levels. First, BRCA2 assists RAD51 to overcome the inhibitory effect of the high-affinity ssDNA-binding replication protein A (RPA), which coats ssDNA and prevents RAD51 loading [44]. Second, BRCA2 guides RAD51 by preferentially binding ssDNA over double-stranded DNA, which is critical for the invasion of a homologous DNA strand and initiation of repair synthesis [44]. When these repair mechanisms fail, it is expected that the cell will activate signals for apoptosis.

An important step in the maintenance of genomic stability is the cell cycle checkpoints, which are tasked with halting the cell cycle progression until it has been ensured that the DNA is intact, therefore maintaining the integrity of the genome from one generation to the next. BRCA1 has been associated with the regulation of both G1-S and G2-M checkpoints.

Following DNA damage, sensors ATM/ATR (ataxia-telangiectasia-mutated/ATM and Rad3-related) phosphorylate BRCA1 at multiple serine residues within the serine cluster domain (SCD). Furthermore, CHK2 kinase, a downstream target of ATM and ATR, also phosphorylates BRCA1 in an ATM-dependent pathway in sites outside the SCD cluster [46,47]. When IR or UV radiation induces DNA damage, the phosphorylation of p53 ser-15 is required to initiate G1-S cell cycle arrest (Figure 1). BRCA1 interaction with BRAD1 is necessary for the ATM/ATR-mediated phosphorylation of p53Ser-15 [48]. Particularly, p53 is needed to activate p21, a cyclin-dependent kinase inhibitor, which induces the G1-S checkpoint arrest. A depletion in the BRCA1–BRAD1 complex jeopardizes the induction of G1-S checkpoint arrest [48].

## 5. BRCA1 Has Also Been Associated with the Regulation of the G2-M Checkpoint

The activation of BRCA1 by the DNA damage sensors has been reported to be critical for the activation of CHK1 kinase, which is in turn essential for the G2-M checkpoint activation following DNA damage. This is achieved by direct interaction between BRCA1 and CHK1 through its BRCT domain [46]. BRCA1 then regulates the Cdc2 kinase Cdc25C and WEE1, which are important cell cycle regulatory proteins that prevent unregulated transition into mitosis [49]. When the cell is arrested at the G2-M cell cycle, it is signaled to induce repair mechanisms or undergo apoptosis if the repairs cannot be accomplished. Cells with defective BRCA1 would bypass the G2-M checkpoint and allow accumulation of DNA damage and subsequent genomic instability. Interestingly, because many cancer cells have defective G1-S checkpoints due to oncogenic transformation, they depend on the G2-M cycle for their survival, especially following exposure to DNA damage. Therefore, focusing on damaging the G2-M checkpoint to induce apoptosis of cancer cells provides a pathway for synthetic lethality.

A key player in determining if the DNA should be repaired or if the cell should undergo apoptosis is tumor protein p53, which acts as a tumor suppressor [50]. When the DNA can be repaired, p53 will activate and recruit other genes to do so, and when the DNA damage is unrepairable, p53 blocks the cell from dividing and signals it to undergo self-destruction [50]. In preventing cells with DNA damage to replicate, p53 helps prevent tumors from developing. Somatic *TP53* mutations occur in most cancers with a rate as high as 50% [51]. This common somatic mutation further exacerbates the genomic instability brought on by *BRCA* mutations and can result in uncontrolled cell division and eventually, cancer.

## 6. BRCA1 Is a Fundamental Protein in Multiple Cellular Processes

In addition to its DNA repair functions, BRCA1 demonstrates a large array of functions in multiple fundamental cellular processes, including cell cycle checkpoint control, ubiquitination, control of gene expression, and chromatin remodeling, which contribute to its major functional role in genomic stability. This diversity of functions provides a framework for developing different treatments for patients with loss of *BRCA1* expression in their tumors [46].

Additionally, BRCA1 plays a critical role in transcription and chromatic remodeling. It is believed that BRCA1 modulates transcriptional regulators though its interactions with transcription factors such as p53, c-Myc, CtIP, ER, and ZBRK1, as well as with the RNA helicase A, a subunit of the RNA polymerase holoenzyme [46,52,53]. Targeting BRCA1 to an amplification region on a mammalian chromosome resulted in localized chromatin de-condensation [53]. Wild-type BRCA1 has also been shown to be required for the transcriptional activation of hGCN5 and TRAAP in a histone acetyltransferase (HAT) complex [53]. Depending on the specific interaction with genes, BRCA1 can function as an up-regulator of tumor suppressors and growth-inhibitor genes or serve as a transcriptional coactivator and corepressor. Mutations in the BRCT domain of the *BRCA1* gene reduce its transcriptional activity.

Another important role of BRCA is ubiquitination, a post-translational modification process that consists of attaching ubiquitin groups to lysine residues in proteins, thus targeting them for proteasome degradation. BRCA1 has E3 ligase activity on its RING-finger domain, which is increased when BRCA1 heterodimerizes with a RING-finger and a BRCT domain [54]. Mutations in the RING domain inactivate BRCA1 E3 ligase activity, which decreases the tumor-suppressor activities of BRCA1. This activity highlights yet another pathway through which *BRCA1* mutations affect genomic stability at several cellular levels.

## 7. Additional Functions of BRCA2

In addition to its important role in DSB repair by HR, BRCA2 safeguards the integrity of DNA by limiting R-loop accumulation, interacting with Smad3 to regulate gene transcription, and playing a role in autophagy and immune system response [55,56,57,58].

R-loops are the result of an RNA:DNA hybrid which displaces a ssDNA, and these are formed when nascent RNA transcripts interact with a complementary DNA structure. Normally, these R-loops are formed at gene promoters and terminators and are regulated and degraded by RNAse H1 [58]. It has been proposed that BRCA2 is recruited by 3′-repair exonuclease 2 (TREX-2) complexes for processing of R-loops [56]. When R-loops accumulate because of a non-functioning BRCA2, they halt the progression of replication forks. This creates an important source of replication stress and cancer-associated instability [59].

BRCA2 is also essential in gene transcription regulation. To do this, BRCA2 forms a complex with Smad3. Smad3 is an important component of the intracellular signaling protein transforming growth factor beta (TGF-beta), a potent inhibitor of tumor cell proliferation [60]. Smad3 possesses transcription activation domains, notably MH1 and MH2, which contribute to the BRCA2–Smad3 complex formation [57]. The BRCA2–Smad3 complex synergizes in the regulation of transcription. BRCA2 co-activates Smad3-dependent transcriptional activation of luciferase transporter and expression of plasminogen activator inhibitor (PAI-1) [57]. Smad3 increases the transcriptional activity of BRCA2 fused to the DNA-binding domain (DBD) of Gal-4, while BRCA2 co-activates DBD-Gal4-Smad3 [57].

A key factor in the immune response of a tumor, autophagy, the process by which a cell consumes its own constituents, has been linked to *BRCA* mutations [58]. In the absence of autophagy, protooncogenic proteins can accumulate and contribute to tumor cell growth, progression though the cell cycle, and angiogenesis. Furthermore, in the absence of autophagy, defective organelles such as mitochondria can accumulate, which results in an increase in reactive oxygen species (ROS) which further damage DNA [60]. Another important function of autophagy is its implication in major histocompatibility complex (MHC), processing a crucial player in the regulation of immune response by allowing the mounting of intracellular material onto MHC class I and II. The elevated expression of MHC-II in breast and ovarian tumors typically associated with *BRCA1* and *BRCA2* mutations has been correlated with better prognosis [61]. There exists evidence to support a negative regulatory role of BRCA2 on autophagy. This is particularly important in the context of PARP inhibitors such as olaparib, whose mechanisms in BRCA1- and BRCA2-deficient cells increase autophagy [62].

*BRCA2* has additional implications in immunity, notably in T cells, which are the primary cell type affected in BRCA2 deficiency in mice [63]. Experiments conducted by Jeong et al. demonstrated that in *BRCA2*-mutant mice, there was a gradual loss of splenic T cell and impaired T cell-dependent immune function. Their study suggested that individuals with a single *BRCA2* allelic mutation could also suffer from T cell deficiency. Furthermore, the study revealed prominent activation of the p53 pathway in BRCA2-deficient T cells, suggesting that in cells lacking functional p53, BRCA2-deficient cells can survive and become tumorigenic [63]. The implication of *BRCA2* mutations on T cell deficiency provides an avenue for immunotherapies.

## 8. Current Available Treatments and Areas of Research

Pancreatic cancer is known to be amongst the deadliest cancers, with a five-year survival of 10.8%. At the time of presentation, 50% of patients have metastatic disease for which the long-term survival is 2% [64]. Even in patients fortunate enough to be diagnosed in the earliest stage of PDAC (stage I), only 20% are able to undergo complete resection (Whipple procedure), after which 41.6% are cured [64]. There is therefore a critical need for continued innovation in treatments. It has been reported that patients with germline mutations tend to perform better in their survival metrics [11]. In the case of hereditary *BRCA* mutations, patients with pancreatic cancers showed almost double the survival rate of those without a hereditary component [11]. Particularly, mutations in DNA damage response and repair (DDR) genes appear to be a positive prognostic factor. Patients with these mutations also demonstrate improved response to certain treatments. This can be attributed to differences in the underlying biology that results from these mutations [41].

There currently exist different types of therapies targeting *BRCA-*mutant PDAC, and numerous studies are ongoing in the hopes of elucidating these mechanisms to improve treatment options.

A.Chemotherapy in *BRCA*-Mutated Cancers

*BRCA*-mutated cancers lack the ability to repair double-stranded breaks induced in their DNA via homologous recombination. It is understood that platinum chemotherapies induce such breaks, thus leading to genomic instability and cell death [65]. Waddell and colleagues confirmed this through *BRCA*-mutated cell lines, finding that they were more susceptible to death induced by DNA damage [66]. This was clinically proven by studying five patients with mutations in *BRCA1*, *BRCA2*, and *PALB2*, and four of these patients had major responses to platinum-based chemotherapy. Emelyanova and colleagues corroborated this finding through the analysis of 543 pancreatic cancer patients. With HRD, specifically *BRCA* and *PALB2* mutations conferred significantly increased survival rates when treated with first-line platinum chemotherapy compared to any other treatment [67].

Currently, platinum-based chemotherapy is routinely administered for treatment in all patients with PDAC. For those patients with good performance status, FOLFIRINOX (5FU, oxaliplatin, and Irinotecan) is recommended [68]. In *BRCA*-mutated borderline resectable PDAC, Golan and colleagues showed increased rates of pathologic complete response and prolonged survival after neoadjuvant FOLFIRINOX [69]. An Italian multicenter analysis confirmed this, showing that three- and four-drug regimens demonstrated a much higher RECIST response than doublet therapies in 85 *BRCA*-mutated PDAC patients (81% and 73% vs. 41% and 56%, respectively) [70]. However, there were more recorded adverse events on these regimens, stressing the need to assess performance status before prescribing this treatment [68]. O’Reilly and colleagues tested cisplatin with gemcitabine in the advanced setting and demonstrated an overall survival of 16.5 months, with very few adverse events [68,71].

Melphalan and chlorambucil are alkylating agents that create intra- and inter-strand crosslinks of tumor DNA. In vitro studies have demonstrated selective toxicity to *BRCA2*-mutated cell lines [72,73,74]. There is currently one clinical trial in BRCA-deficient patients testing melphalan combined with infusions of Vitamin B12, BCNU, Vitamin C, and Hematopoietic stem cells [75]. Unfortunately, melphalan’s use is limited due to its toxic profile and difficult tolerability. Chlorambucil’s anti-tumor efficacy was shown to be similar to cisplatin and is being tested in BRCA-deficient PDAC. This study is estimated to be completed in December 2023 [76].

B.PARP Inhibitors in *BRCA*-Mutated Cancers

Cells deficient in homologous recombination, especially *BRCA*-mutants, will shunt to alternative DNA repair pathways. BRCA-deficient cancers often rely on PARP, a protein primarily used in the repair of single-stranded breaks. When these cancers are treated with chemotherapy, surviving cells rely on the PARP DNA repair pathway to fix the damaged DNA, and thus resist chemotherapy [77]. In these cancers, drug-induced PARP inhibition will lead to two non-functional DNA damage repair pathways, causing “synthetic lethality” that results in either necrosis or apoptosis of the cell [78]. DNA damage is more frequent in rapidly dividing cells, so synthetic lethality is more likely to occur in tumor cells while sparing healthy tissue. As such, PARP inhibitors have been investigated heavily for use in *BRCA*-mutated PDAC (Figure 2).

The primary PARP inhibitor recommended by the National Comprehensive Cancer Network (NCCN) for *BRCA*-mutated pancreatic cancer is olaparib [68]. Currently, olaparib is indicated for metastatic prostate, pancreatic, and breast cancer, as well as advanced ovarian cancer [79]. The POLO trial tested the use of olaparib vs. placebo for maintenance therapy in metastatic PDAC after at least 16 weeks using platinum-based chemotherapy. Progression-free survival in the olaparib group was nearly double that of the placebo group, at 7.4 months, compared to 3.8 months [80]. Although the difference in overall survival was not significant, it has been added to the NCCN guidelines [69]. Olaparib has also been shown to have anti-tumor effects on mutations with a BRCAness phenotype. A two-center study in the US and Israel demonstrated partial response and stable disease in 13 of 23 patients with *BRCAness* [81].

Rucaparib is another PARP inhibitor that is currently approved for ovarian and prostate cancer, tested as maintenance therapy in BRCA mutants, and demonstrated a 13.1 month PFS [82]. To place this finding in perspective, the PFS in the rucaparib trial (13 months) exceeds the OS estimates from the landmark Phase III trials of FOLFIRINOX (11 months) or Gemcitabine plus nab-paclitaxel (8 months) as a first-line treatment of metastatic pancreatic cancer. This highlights not only a PARPi benefit in the BRCA mutant pancreas cancer population, but also the relatively favorable outcome of BRCA mutant PDAC compared to non-*BRCA**-*mutant PDAC. Interestingly, responses with rucaparib seem to be varied. A study of rucaparib following one or two lines of platinum-based chemotherapy was closed due to a lack of initial response in the first fifteen patients enrolled; however, the final analysis showed very promising data. The final four patients enrolled in the study had two partial responses, one confirmed complete response, and one unconfirmed complete response [83]. These data show the potential efficacy of rucaparib and warrant its further investigation to elucidate the mechanisms of resistance.

Veliparib is another PARP inhibitor that recently showed promise in advanced *BRCA*-mutated ovarian cancer when combined with first-line chemotherapy, significantly increasing PFS [84]. However, when tested in PDAC alone, among 16 patients, there was only 1 unconfirmed partial response, 4 patients (25%) with stable disease, and the remainder with disease progression. Additionally, six (38%) patients experienced grade III toxicity [85]. While most PARP inhibitors directly limit the PARP proteins from functioning, talazoparib has an additional function of PARP trapping on the surface of a DNA single-stranded break, making it approximately 100 times more efficacious. It is pharmacologically active at much lower concentrations than standard PARP inhibitors, with an IC_50_ of 5 nmol/L [86,87]. Talazoparib is currently approved for treatment only for *BRCA*-mutated breast cancer [79]. A phase I clinical trial by Bono and colleagues demonstrated that talazoparib had pharmacological activity against various *BRCA*-mutated cancers, including pancreatic cancer. While caution should be used when interpreting response rates from this phase I trial for which primary endpoints relate to safety/toxicity, it is notable that among ten patients with metastatic PDAC, two of the ten had partial response (overall response rate of 20%), and one had stable disease (disease control rate of 30%) [88]. Additionally, the authors are presently investigating the role of talazoparib treatment in the neoadjuvant setting, with the goal of increasing the number of patients who complete surgical resection (Table 1).

AZD 5305 is a novel, highly selective PARP inhibitor currently being tested in the ongoing phase I/IIa PETRA trial (NCT04644068). AZD 5305 is a highly potent PARP1 inhibitor that showed no PARP2 activity, which may limit its gastrointestinal and hematologic side effects. Like Talazoparib, AZD 5305 also has significant PARP1 DNA-trapping activity. Of 40 evaluable patients, 10 had partial responses, 11 patients had stable disease, and 19 patients had progressive disease. The most common adverse events of any grade were nausea in 34.4% and anemia in 21.3% of patients [89].

**Table 1 cancers-14-02453-t001:** Recent advances in PARP inhibitor therapy, adapted from Gupta and colleagues [90].

Investigators	Phase	Patient Population	Number of PDAC Patients	Intervention	Outcome	Ref
Kauffman et al.	II	PDAC with gBRCA1/2 mutation following progression on gemcitabine	23	Olaparib 400mg PO BID	ORR 22% PFS 4.6 months OS 9.8 months	[91]
Shroff et al.	II	PDAC with any BRCA mutation, previously treated with 1-2 lines	19	Rucaparib 600mg PO BID	ORR 16%	[83]
Lowery et al.	II	PDAC with gBRCA mutation or PALB2 mutation, 1-2 prior lines of treatment	16	Veliparib PO BID PO	PFS 1.7 months OS 3.1 months	[85]
Golan et al.	II	PDAC with BRCA-appearing phenotype, first or second line	32	Olaparib PO BID	PFS 14 weeks in Israel 25 weeks in the US	[81]
Golan et al.	III	PDAC with gBRCA mutation that has not progressed on firstline platinum-based treatment	92 olaparib 62 placebo	3:2 olaparib versus placebo	ORR 37%	[80]
Reiss et al.	II	PDAC with g or s BRCA or PALB2 mutations that has not progressed on firstline platinum-based treatment	24	Rucaparib 600mg PO BID	ORR 37%	[82]
Chiorean et al.	II	PDAC including g or s BRCA or PALB2 mutations	108	1:1 veliparib + FOLFIRI versus FOLFIRI alone	OS 5.1 vs 5.9 months PFS 2 months vs 3 months	[92]
Pishvaian et al	I/II	PDAC with g or S BRCA or PALB2 mutations or relevant breast or ovarian family history	22	Veliparib + mFOLFOX6	OS 8.5 months PFS 3.7 months	[93]

C.PARP Inhibitors in Combination with Other Agents

PARP inhibitors have also been tested in the combination setting in *BRCA*-mutated PDAC. In 2020, O’Reilly and colleagues tested the effect of gemcitabine and cisplatin with and without Veliparib. They showed that although the response rate of the triple therapy was slightly better (74.1% vs. 65.2%), this did not reach statistical significance. The veliparib arm showed decreased PFS as well as significantly increased adverse events, and therefore gemcitabine and cisplatin alone remain a standard treatment in this setting [6,71].

While PDAC is notoriously immunologically cold, PARP inhibitors have been demonstrated to increase PD-L1 expression, stimulating the immune system’s ability to recognize and attack tumor cells [94]. In addition, it has been shown that neoantigen formation is increased by mutations, and PARP inhibitors play a large role in driving this process [94]. Investigators are searching for ways to engineer T cell therapies to directly attack mutations in the US National Clinical Trials Network-based cancer genome [95]. PARP inhibitors are being investigated in the SWOG S2001 trial combining olaparib with immune checkpoint/PD-1 inhibitor pembrolizumab in the maintenance setting [96]. NCT04548752 is currently recruiting for this investigation [97]. Another study is currently investigating niraparib and the PD-1 inhibitor dostarlimab under similar mechanisms of action, projected to be completed in December 2022 [98].

Retrospective reviews of ipilimumab and nivolumab treatment showed excellent responses in patients with HRD PDAC. Of 11 patients with PDAC or ampullary carcinoma on this treatment, 3 patients had complete responses, 1 had a partial response, and 2 had stable disease [99]. This demonstrates promise for the future use of combination immunotherapies in these patients. Additionally, it could prove a fruitful area of study to test these therapies in combination with PARP inhibitors.

Wang and colleagues are currently conducting a study of a novel Bromodomain and extra-terminal (BET) inhibitor AZD5153 in combination with olaparib (NCT03205176) [100]. BET inhibitors are described to inhibit transcriptional complexes at bromodomains that are necessary for tumor survival, and preclinical models show synergy between PARP and BET inhibitors. This combination is currently being trialed in humans, and preliminary phase I data reported that AZD5153 monotherapy was safe [101].

Liu et al. demonstrated an increased PFS in *BRCA**-*mutated ovarian cancer patients treated with olaparib in combination with cediranib, a VEGF inhibitor [102]. This combination is currently being investigated in patients with pancreatic cancer as well (NCT02498613) [103].

D.Other Novel Treatments

Many patients will develop resistance to PARP inhibitors altogether. The most commonly described mechanisms include reversion to *BRCA* wild-type status, overexpression of SLFN11, and 53BP1 loss [104,105,106]. In vitro studies have also shown this resistance co-occurring with an upregulated ATR/CHK1 pathway. This indicates that cells may be relying on this pathway for substituted DNA repair [107]. The CAPRI trial is currently investigating a novel ATR inhibitor, AZD6738, in combination with olaparib in recurrent ovarian cancer, some of whom have been pretreated with PARP inhibitors [108].

WEE1 kinases are important regulators of the cell cycle, arresting a cell with DNA damage at the G2M checkpoint so the DNA damage can be repaired. If WEE1 kinase is inhibited, the cell will be forced into mitosis regardless of DNA damage, and thus will produce genomically unstable cells. Thus, WEE1 kinase inhibitors are salient partners for platinum chemotherapy [109]. A study by Hartman and colleagues analyzed a WEE1 inhibitor in combination with various types of chemotherapy in PDX models with PDAC. In cell lines harboring somatic *TP53* mutations, inhibiting WEE1 in combination with irinotecan or capecitabine demonstrated significant inhibition of tumor growth, while the same results were not seen in cell lines with wild-type *TP53* [110]. Cuneo and colleagues investigated adavosertib, a WEE1 inhibitor, as a treatment for locally advanced PDAC with *BRCA* mutations in combination with gemcitabine and radiation. Reported overall survival was 21.7 months, while PFS was 9.4 months [111]. These data endorse the potential clinical utility of WEE1 kinase inhibitors.

E.Future Research Endeavors

Despite the prognostic and predictive relevance of *BRCA* testing in pancreatic cancer, adoption of universal germline genetic testing has been slow. There is presently a paucity of relevant data describing its adoption in the literature, so results are expected to emerge first from institutional data. To that end, the authors have launched a formal institutional quality improvement initiative to uncover testing adherence as a first step in understanding potential barriers to adoption. As testing becomes more widespread, it is possible that we will learn more about variants of unknown significance (VUS), which currently make up 44.4% of all identified *BRCA* mutations [112]. Of the variants that are reclassified, 91.2% are downgraded to benign classifications [113]. However, this is only the case of 7.7% of VUS, with the remaining ones largely unstudied. As mutational testing becomes more widely available, there will be more data to explore with preclinical models and clinical research.

Preclinical studies of *BRCA* mutations have many potential therapeutic targets that have been demonstrated in clinical trials. However, studying specific gene knockouts is exceptionally time- and resource-intensive. CAPAN1 is a *BRCA2*-mutated cell line commercially available for experiments as a germline model. However, the lack of additional germline models limits the ability to conduct a breadth of experiments. With the advent of CRISPR-cas9 gene editing, researchers have successfully induced specific BRCA loss of function variants in cell lines, allowing these specific mutations to be studied in vitro with much less time investment, driving forward research [114]. Additionally, the authors are undertaking the development of germline organelle model systems.

Historically, PARP inhibitor therapies have been targeted to germline BRCA-mutant patients. However, recent studies have indicated that patients with somatic *BRCA* mutations might benefit equally from these treatments, and therefore somatic profiling of PDAC patients is also beneficial [104,115]. Several phase II trials for PARP inhibitors, including Niraparib (NCT03601923), Rucaparib (NCT04171700), and Talazoparib (NCT04550494), are now recruiting patients with somatic *BRCA* mutations [116,117,118]. Preliminary data on the Rucaparib trial suggest efficacy in patients with somatic BRCA2 mutations, which supports that a larger population of PDAC patients could benefit from PARP inhibition. Further characterization of patients receiving PARP inhibitors is nonetheless needed to better understand how this subset of patients respond to therapy.

## 9. Conclusions

The discovery of *BRCA* genes in the 1990s created a field of research to better understand the mechanisms and function of these proteins in the hopes of treating breast and ovarian cancer. Since their discovery, these tumor-suppressor genes have also been implicated in pancreatic cancers. Although only 10% of PDAC appear to have a familial hereditary component, patients with *BRCA* mutations appear to have better prognosis. The complexity of these genes and their numerous domains allow for a multitude of protein interactions, which make these genes prime targets for cancer therapies. BRCA1 and BRCA2 play a major role in the HR repair mechanism of DSB. In addition to this, both BRCA1 and BRCA2 are implicated in multiple cellular processes, ranging from cell cycle checkpoint to autophagy [118]. This predisposes *BRCA* mutant cancers to several ways of achieving synthetic lethality.

Novel therapies including next-generation PARP inhibitors such as AZD 5305, which show more selective targeting, may prove to be more efficacious and more tolerable. Unique targets such as WEE1, an important regulator of the cell cycle, may lead to synergy with both traditional chemotherapy as well as new targeted agents and PARP inhibitors in the future. These combinations may also be utilized to overcome PARP inhibitor resistance. Inclusion of somatic *BRCA* mutations in clinical trials may expand the number of patients who may undergo targeted therapy. As a result, potential treatment paradigms are vast, with chemotherapy sensitivity and targeted treatment pairings as promising future avenues. As such, there are many opportunities for continued research in the field.

## Figures and Tables

**Figure 1 cancers-14-02453-f001:**
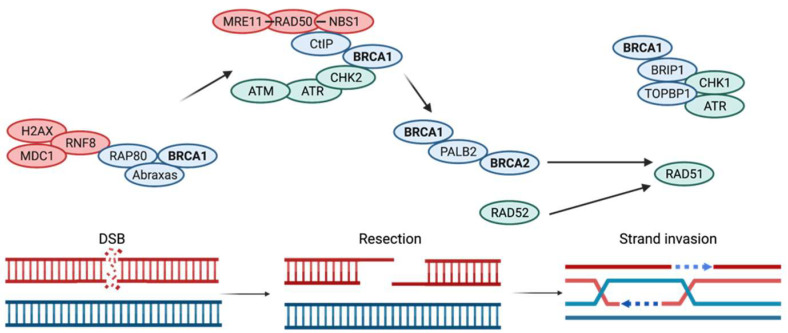
Molecular mechanisms involved in the double-stranded break repair by homologous recombination (HR).

**Figure 2 cancers-14-02453-f002:**
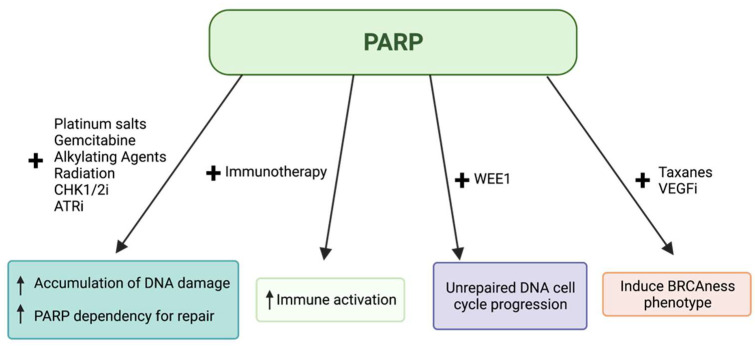
Overview of the mechanisms of PARP inhibitors in combination with other agents.

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
