# Peer review of "BRCA-Mutated Pancreatic Cancer: From Discovery to Novel Treatment Paradigms"

_cancers, 2022, doi:10.3390/cancers14102453_

Round 1

Reviewer 1 Report

Please, see the attachment, thanks!

Reviewer 2 Report

In this review, Authors explore the critical functions of BRCA1 and BRCA2 in the repair mechanisms and genome stability  and the implication for therapy.

I think the work is well written and suitable for publication with some revisions.

I suggest to add epidemiology of  BRCA mutations and "BRCAness"phenotype in PDAC: now are available some evidences about the real impact of HRD in PDAC, also in some "real life" experiences. 

I suggest also to explore the clinical significance of VUS of BRCA1 and BRCA2 and the explorative evaluation of somatic mutation of these genes for therapy.

Reviewer 3 Report

Cancers (Manuscript Number: ISSN 2072-6694)

BRCA Mutated Pancreatic Cancer: From Discovery to Novel Treatment Paradigms

This paper discusses the BRCA1/2 mutation events in pancreatic cancer. The author introduced the discovery history of BRCA1/2 gene from breast cancer to pancreatic cancer. Other than the basic function involving in DNA repair pathway, the author described the other functions of BRCA1/2, such as cell cycle, ubiquitination, autophagy and even immune system response, which could be interesting. Based on the conventional and unconventional function of BRCA1/2 genes, the author listed the mainstream of treatments stratified by genomic information including platinum, PAPRi and other novel combined therapies.

  1. Misleading Titles

Testing for deleterious germline mutations mainly not the somatic mutation inBRCA1/2impacts patient selection for platinum-based chemotherapy regimens and selection of patients who are candidates to receive maintenance therapy with olaparib.I would recommend the author specify the gBRCA1/2 other than BRCA1/2. If the author wants to describe the all mutation types of BRCA1/2, they should add the explanation of the difference between germline and somatic mutations in treatment stratification in pancreatic cancer.

Somatic BRCA mutation, presenting in 3% to 4% of PDAC, is an important subset of PDAC. Since the authors reviewed “BRCA Mutated Pancreatic Cancer”, somatic mutation might be necessary. Moreover, the benefit to platinum and PARPi therapy of somatic BRCA mutation in PDAC remains contradictory. This is supposed to be taken into account. The authors may review the following and more papers:

  • 2021 Dec 1;127(23):4393-4402. doi: 10.1002/cncr.33812
  • JCO Precis Oncol. 2018;2018:PO.17.00316. doi: 10.1200/PO.17.00316
  • 2021 May;160(6):2119-2132.e9. doi: 10.1053/j.gastro.2021.01.220
  1. Careless keywords

In am so confused about the relationship of keywords (hedging; transaction costs; dynamic programming; risk management; post-decision state variable) with this paper

  1. Illogical subtitles

The author described main three parts of this paper: discovery, function and treatment. The present subtitles are not well organized. I would recommend as follows:

1.Discovery

1.1Discovery in breast cancer

1.2 Discovery in ovarian cancer

1.2Discovery in pancreatic cancer

2.Funcion

2.1 DNA repair pathway

2.2 Cell cycle

  • authophy

3.Treatment

3.1 Platinum treatment

3.2 PARPi

3.3 Other novel treatment

  1. Low quality of the figure

Good figure could help us remember some concepts of good storytelling. But all the figures in this manuscript adapted from other literature are not well organized and impressive.

  1. Limited current results

This is a very limited/poor overview of the current state-of-play in terms of PARPi and novel treatments. There are a lot of ongoing clinical trials about the combination therapies with PARPi such as VEGFi (NCT02498613), immune checkpoint inhibitor (NCT04548752,NCT04753879 and NCT03404960) and BETi (NCT03205176). The future and ongoing promising clinical trials would be interesting and novel to the readers.

  1. Inaccurate abstract

In abstract part, the author mentioned CDKN2A is quite confusing. And in the results section, the author declared “BRCA1 has also been associated with the regulation of the G2-M checkpoint” and “BRCA1 is a Fundamental Protein in Multiple Cellular Processes” not BRCA2. However, in abstract part, the author described both BRCA1 and BRCA2 played fundamental roles in cell cycle checkpoint control, ubiquitination, control of gene expression, chromatin remodeling, autophagy and even immune system response. And from the title, it related to treatment. In abstract, the reader would be distracted by the prognostic and predictive function of BRCA1 and BRCA2.

  1. In line 365-372, as the authors mentioned, FOLFIRINOX or gemcitabine plus cisplatin are recommended as first line treatment for adjuvant and metastatic BRCA-mutated PDAC. They cited a study which compared FOLFIRINOX versus gemcitabine alone in metastatic PDAC (not BRCA- mutated). How could they make a conclusion that “FOLFIRINOX is recommended as a first line, aggressive treatment in metastatic PDAC patients with good performance status”?

I think the authors attend to demonstrate that BRCA-mutated PDAC patients might benefit to platinum, but with an inappropriate example.

AND what about gemcitabine plus cisplatin in BRCA-mutated PDAC patients?

  1. The third part of the paper “3. Predisposition to Pancreatic Cancer with BRCA and Associated Gene Mutations”. The authors should focus on BRCA rather than other associated genes, and it takes too long.

Reviewer 4 Report

Although approach over the 7 first sections is, without doubts, outstanding and brilliant I miss the same quality on wording from section 8 onwards, maybe explained by first author background. I would highly recommend to rewrite entirely the first paragraph of this section (mainly regarding description of 5y cumulative OS, % of resection in stage 1 PDAC etc).

I would also recommend to include some of the pletora of studies that have analized role of chemo in BRCA mutated subjects and also a deeper review of recent articles focused in this particular scenario, toxicity, efficacy etc (PMID: 32314163, 35309086,  34392104, 33551067, 31542591) but mainly the follow one: 31976786 that should be described in detail. On the other hand, role of melphalan, chlorambucil could be described shorter or even removed.

Round 2

Reviewer 1 Report

The authors have greatly improved the manuscript according to the comments. The manuscript can be accepted for publication in the Special Issue "Aberrant Signaling Pathways in Pancreatic Cancer" of "Cancers" .
